# Difficult Doctor–Patient Relationships and Attachment Style in Primary Care

**DOI:** 10.3390/healthcare13222952

**Published:** 2025-11-18

**Authors:** John H. Porcerelli, Reem Eissa, Pierre Morris

**Affiliations:** 1Department of Psychology, University of Detroit Mercy, Detroit, MI 48221, USA; 2Department of Psychiatry, University of Texas Health, Houston, TX 77030, USA; eissareem@gmail.com; 3Department of Family Medicine, Oakland University William Beaumont School of Medicine, Rochester, MI 48309, USA; pierremorris@oakland.edu

**Keywords:** primary care, attachment, health, difficult doctor-patient relationship

## Abstract

**Background and Objective:** Physicians experience 10–30% of patient interactions as being interpersonally difficult, but the sources of difficulty are not well understood, despite the frequency of anecdotal studies of the “difficult patient.” This study examined whether patients rated as “difficult” by their physicians have more of an insecure attachment style, and whether patients not rated as “difficult” have more of a secure attachment style. **Methods:** The participants were 100 primary care patients and 28 physicians from a university-based family medicine residency training program in a suburb of Detroit, Michigan, USA. **Results:** The results revealed that, while adjusting for the patients’ age, education, and income, fearful attachment and the physician’s rating of the patient’s overall health were significantly associated with the physician’s perception of patient difficulty. **Conclusions:** This study suggests that residency training should include behavioral science education in the assessment and management of insecure attachment styles to ensure that such patients receive optimal care and support the mission of primary care medicine.

## 1. Introduction

Physicians experience 10–30 percent of interactions with patients as being interpersonally difficult [1,2,3,4]. Prevalent in primary care settings, difficult patients are shown to experience more functional impairment and less satisfaction with provision of care, suffer more psychiatric illnesses, and utilize healthcare services at a higher rate compared to their non-difficult counterparts. In an era of prioritizing patient-centered care, research examining physician–patient interaction is limited despite the collaborative relationship being viewed as a vehicle for patient wellness and a pathway for improving population health. Understanding interpersonal aspects of such relationships would inform implications on medical outcomes and healthcare utilization patterns in primary care. Moreover, using an established model of interpersonal interactions may be useful in understanding why certain patients are experienced as more difficult to work with by their physicians than others.

Attachment theory proposes that based on the quality of care received in early relationships, infants develop expectations about whether or not others can be relied on for comfort at times of distress [5]. Mental schemas (also referred to as models of self-and-other) formed through these early experiences with caregivers influence how individuals experience and engage in close relationships, particularly to the degree to which they perceive others as responsive and reliable. These models of self-and-other are activated in interpersonal settings, particularly those where one is in need of comfort [6]. Exploring the relationship between attachment style and the physician–patient relationship rests on the premise that a healthcare provider serves an attachment function in the context of illness, a circumstance that is thought to activate the attachment system. Adult attachment styles include secure, preoccupied, dismissing, and fearful [5]. Adults whose attachment style is predominantly secure are believed to have experienced consistent early caregiving [7], leading to the development of positive models of self and other. These individuals are able to depend on and be comforted by others, such as healthcare providers in the context of illness.

Inconsistently responsive early caregiving is posited to lead to the development of a preoccupied attachment style [8]. These individuals are often emotionally dependent on the approval of others, and generally have poor self-worth, consistent with a negative model of the self. They tend to exaggerate their needs for comfort to evoke support from caregivers. As patients, they may amplify their physical symptoms, lack confidence in the self-management of their illness, and may seek care frequently for minor concerns. Their physicians may experience them as dependent, needy, and solicitous of care [9].

Individuals with dismissive or fearful attachment styles generally avoid support-seeking and deny attachment needs [10]. Adults whose attachment style is predominantly dismissive are believed to have experienced emotionally unresponsive early caregiving, leading to a development of “compulsive self-reliance” [11] and a discomfort with trusting others. As patients, they may minimize or deny their need to be comforted, downplay their symptoms, and remain disengaged with their physician [12]. Doctors may experience them as undemanding and unproblematic; this may lead to provision of suboptimal care in medical settings with high practice volumes [9].

For adults with a primarily fearful attachment style, their desire for social contact is inhibited by fear of rejection [8]. This fear is thought to have developed through contact with harsh and rejecting caregivers, leading to a negative model of both self and others, and a tendency to vacillate between approach and avoidance behaviors. Physicians typically experience these patients as frustrating, alternating between demanding immediate treatment and failing to adhere to the physician’s treatment attempts. Patients with dismissing and fearful attachment styles have been shown to report low levels of trust and difficulty building longstanding collaborative relationships with healthcare providers [9].

Researchers have explored how adult attachment styles may influence healthcare experiences. For example, a study examining the interpersonal experience of patients with diabetes within the healthcare system revealed that patients find rushed and impersonal communications with their physicians to be frustrating [12]. In addition to these factors, patients with fearful and dismissive attachment styles noted that the perceived “wall” between healthcare providers and patients often interferes with their ability to trust and engage with their physicians. In a recent study of attachment style and the doctor–patient relationship in palliative care patients, securely attached patients developed significantly better relationships with their doctors than did insecurely attached patients [13]. There were no differences detected in the quality of the doctor–patient relationship among preoccupied, dismissive, and fearful patients.

A narrative review of the literature revealed that research on attachment style led to contributions in the understanding of physician–patient interaction and medical outcomes in the treatment of chronic conditions [14,15], such as diabetes [16]. Within an insecure attachment group, patients with dismissing attachment styles had lower attendance rates, and those with preoccupied and fearful attachment styles had higher attendance rates, than patients with secure attachment styles in a female sample of primary care patients [12]. This may suggest that attachment anxiety leads to greater attendance, a care-seeking behavior.

Other correlates to attachment include medication adherence and mortality rates. Dismissing attachment style was found to correlate with lower medication adherence and to mediate the relationship between depression and medication treatment adherence by exacerbating the negative relationship [14]. This is thought to be due to the self-sufficiency, low desire for collaboration with the physician, higher likelihood of missed appointments [9], or mistrust of professionals and rejection of medical advice [12] than individuals with a dismissing attachment style are likely to demonstrate. These patients do not often interact with healthcare providers; this underutilization may help explain the significant rates of mortality in this group compared to other attachment groups [12]. In a 5-year prospective study of 3500 diabetic patients, those with a dismissing attachment style were found to have a 33% increase in risk of death by all causes relative to their securely attached counterparts [17].

This study will test this prediction in a primary care setting where patients are expected to partner with providers for longstanding healthcare relationships. We hypothesize that patients rated as “difficult” by their physicians have more of an insecure attachment style, and patients not rated as “difficult” tend to have more of a secure attachment style.

## 2. Materials and Methods

### 2.1. Participants and Procedure

Participants included primary care patients from a university-based family medicine residency training clinic in a suburb of Detroit, Michigan, USA. A total of 175 consecutive primary care patients were given an information sheet containing details about the research project upon entering the clinic waiting room. A total of 100 patients (57%) agreed to participate in this study. Participants completed measures in the waiting room prior to their appointments with their physician. Patients completed self-report measures about their physical health, mental health, and attachment style. Primary care physicians (n = 28) completed overall health ratings of their patient and ratings of their experience of the interaction with the patient following the medical encounter. Physicians were blind to the patient’s attachment classification. Informed consent was obtained from both patients and physicians who participated in the study. There were no missing data—participant data were complete. The data for this study was originally collected in 2016 and is part of an archival data set. Data on attachment styles were collected at that time but has not been used in any prior study. Our research team is embarking on a large-scale study of attachment and doctor–patient relationships and is using this 2016 data set as a pilot project. The rationale for using archival data is based on the fact that there are so few studies on attachment and doctor–patient relationship in the literature and that this attachment data has not been used in any prior investigation. The original study was approved by the Wayne State University Institutional Review Board. Patients received a USD 10 honorarium for their participation.

### 2.2. Measures

Relationship Questionnaire (RQ): A 5-item instrument based on Bartholomew and Horowitz’s [18] four-category model of attachment. The RQ has adequate convergent validity with other attachment measures [18]. Participants chose from brief paragraphs describing the four attachment styles which best describes their relationship style. Participants then rated their degree of correspondence to each prototype on a 7-point scale (1 = not at all like me; 7 = very much like me). Similar to the definition provided by Hazan and Shaver [19], the secure prototype is described by a comfort relying on others and being relied on, a comfort with emotional closeness, and the absence of concerns regarded being accepted by others. The fearful prototype is described by a desire for emotional closeness with a difficulty trusting and depending on others, and a fear of being hurt by them. The preoccupied prototype describes an individual uncomfortable without close relationships, wishing to be emotionally intimate with others, and fearing that others will not reciprocate. Finally, the dismissing prototype is described by a comfort without close relationships and valuing one’s independence and self-sufficiency. The RQ has been demonstrated to have good convergent and discriminant validity [20] and predictive validity [21].

Difficult Doctor–Patient Relationship Questionnaire [1]. The DDPRQ-10 is a 10-item measure for identifying physicians’ perception of difficult patient encounters. The DDPRQ-10 was constructed from the original 30-item DDPRQ [21]. The DDPRQ-10 is a reliable and valid measure for primary care settings [4,22,23]. DDPRQ-10 items include the following: “How much are you looking forward to this patient’s next visit after seeing this patient today?”, “How frustrating do you find this patient?”, “How manipulative is this patient?”, “To what extent are you frustrated by this patient’s vague complaints?”, “How self-destructive is this patient?”, “Do you find yourself secretly hoping that this patient will not return?”, “How at ease did you feel when you were with this patient today?” “How time consuming is caring for this patient?”, “How enthusiastic do you feel about caring for this patient?”, and “How difficult is it to communicate with this patient?” Items are rated using a six-point scale with values ranging from 1 (not at all) to 6 (a great deal). A cut-off score of 30 or greater is used to identify difficult physician–patient encounters [21]. The internal consistency for this study was 0.89.

Patient Health Status Visual Analogue Scale. The VAS was completed by doctors to indicate, on a scale of 1 to 100, the patient’s level of overall health. VASs have shown adequate convergent validity with well-validated measures of overall health and health perception [24]. Higher scores represent better health.

## 3. Results

Participant ages ranged from 18 to 73 years (M = 40, SD = 15). Seventy percent were female, 63% were White, 29% were Black, and 8% were “Other”. Forty percent had high school education or less, 31% had some college, and 29% had a college degree. A proportion of 50% were single, 39% were married, and 11% were divorced or separated. Sixty-six percent had an annual income of less than USD 40,000. A total of 16% of patients were rated by their physicians as “difficult” (i.e., a total DDPRQ-10 scale score of 30 or greater).

According to Curran et al. [25], variables are relatively evenly distributed if skewness is less than 2.0 and kurtosis is less than 7.0. Thus, all variables in this study were relatively evenly distributed.

Statistical analyses for this study included correlations of study and demographic variables (Table 1). Age, education, and income were found to correlate with several study variables. Therefore, age, education, and income were included as covariates in multivariate analysis. Study variable intercorrelations are reported in Table 2. Secure attachment was negatively correlated with the other three attachment variables. Fearful, preoccupied, and dismissive attachment styles were all positively correlated with one another. Secure and fearful attachment styles and physician-rated patient health were significantly correlated with DDPRQ-10 total scores.

A multivariate analysis of covariance (MANCOVA) was computed with patient difficulty (yes/no) as the independent variable and attachment styles and physician-rated patient health as dependent variables. Patient age, education, and income were included as covariates (Table 3). The MANOVA Wilk’s Lambda result was significant, F(9,91) = 4.05, *p* = 0.002. Box’s M test indicated that the assumption of homogeneity of variance–covariance matrices was met, M = 10.53, *p* = 0.49. Although means for all independent variables were in the expected direction, only fearful attachment and physician-rated patient health were significantly different. That is, fearful attachment means were greater among patients rated as difficult by their physicians as compared to patients rated as not difficult. Patients rated as difficult were also rated as having poorer health than patients not rated as difficult.

## 4. Discussion

This study assessed the relationship between the physician’s perception of a patient as “difficult” and the patient’s attachment style. The results revealed that, while adjusting for age, education, and income, fearful attachment and physician-rated overall health were significantly associated with the physician’s perception of patient difficulty. Thus, patients with higher ratings of fearful attachment and worse health were rated as more difficult to work with than patients with secure, preoccupied, and dismissive attachment styles.

Fearfully attached patients tend to demand immediate treatment from their physicians but often fail to adhere to the physician’s treatment attempts [10]. This vacillation can pose a threat to the health of an already sick patient, leading the doctor to worry about the patient or to experience the patient as noncompliant, self-destructive, and/or frustrating. Our results show that the most difficult patients for physicians in this study, the majority of whom are in residency training, are sicker patients who solicit care but fail to trust themselves and the doctor in the management of their illness. Residents may also deflect their feelings of incompetence onto patients by experiencing them as difficult.

These findings also point to a potential public health problem. Research indicates that insecure attachment styles are associated with dysregulated physiological responses to stress, risky health behaviors, susceptibility to physical illness, and poorer disease outcomes [15,26,27]. Our findings suggest that individuals with fearful attachment styles may pose a challenge to primary care physicians, especially those in residency. Primary care physicians who experience encounters with these patients as negative, run the risk of implicitly or explicitly conveying their negative feelings to such patients and thus increase the likelihood that the patient will not follow the provider’s recommendations or may avoid future medical encounters altogether [26]. It would be important to determine the prevalence of patients who avoid care following difficult encounters to underscore the public health implications of such encounters.

Future research could consider assessing attachment style in both patients and physicians, especially because insecure attachment styles are as prevalent in medical students as they are in the general population [28]. A recent review article reported that physicians with insecure attachment styles are likely to struggle with important aspects of mental functioning that can contribute to errors in medical decision making and diagnostic errors [29]. Future studies could include observer-ratings of doctor–patient encounters to understand physicians’ contribution to experiencing encounters as difficult.

Although psychosocial and behavioral science training have made important contributions to the development of family medicine over the past 50 years [30], training in the assessment and management of attachment styles and pathology is nearly absent from their curricula. Therefore, we recommend behavioral science training for both faculty and residents alike, include didactics as well as objective structured clinical exam (OSCE [31]) training focusing on diagnosis and management of secure and insecure attachment styles.

A limitation of this study is that difficult patients constituted only 16% of our sample, limiting the study’s ability to detect differences in patient difficulty between attachment groups. However, this statistic aligns with the literature where 10–30% of patient interactions are cited as being interpersonally difficult [1,2,3,4,22]. Future studies should employ patient feedback on their interactions with physicians to better understand their experience of the encounter. Despite these limitations, study results provide clinically relevant knowledge that can be applied to a patient-centered approach and incorporated into residency training and education (e.g., patient-centered clinical medicine skill training [32] and unconscious bias mitigation strategies [33]), to improve patient care with more challenging patients and to improve the overall public health of this vulnerable patient population. Lastly, the use of archival data is a limitation of the study because medical encounters in 2016 may be qualitatively different than encounters in 2025 due to the increased use of digital technology by healthcare providers during such encounters [34].

## 5. Conclusions

Attachment style is an important construct for physicians to understand in order to provide optimal care to those patients who suffer from a fearful attachment style. Knowing how to assess and manage encounters with such patients may improve the physician–patient relationship, improve patient compliance, and perhaps result in better health outcomes.

## Figures and Tables

**Table 1 healthcare-13-02952-t001:** Correlations of study variables and demographics (N = 100).

	Secure	Fearful	Preoccupied	Dismissive	Patient Health	DDPRQ-10
Age	0.04	−0.31 **	−0.11	−0.05	−0.13	−0.13
Race	−0.19	−0.04	−0.01	0.16	−0.09	0.13
Gender	0.05	0.10	0.09	0.09	−0.03	0.16
Education	0.22 *	−0.16	−0.26 **	0.12	0.17	−0.15
Income	0.33 ***	−0.37 ***	−0.27 **	−0.08	0.27 **	−0.35 ***

Note: * r ≥ 0.05 level; ** r ≥ 0.01 level; *** r ≥ 0.001 level. Race was dichotomized to White and Other. DDPRQ-10 is the total score. Gender = male and female.

**Table 2 healthcare-13-02952-t002:** Intercorrelations of study variables (N = 100).

	Secure	Fearful	Preoccupied	Dismissive	Patient Health
Secure	--	--	--	--	--
Fearful	−0.55 ***	--	--	--	--
Preoccupied	−0.32 ***	0.46 ***	--	--	--
Dismissive	−0.17	0.23 *	−0.03	--	--
Patient Health	0.19	−0.04	0.07	−0.12	--
DDPRQ-10	−0.26 **	0.31 ***	0.11	0.06	−0.45 ***

Note: * r ≥ 0.05 level; ** r ≥ 0.01 level; *** r ≥ 0.001 level. Race was dichotomized to White and Other. DDPRQ-10 is the total score. Gender = male and female.

**Table 3 healthcare-13-02952-t003:** Difficult doctor–patient relationships and attachment style.

	Non-Difficult N = 84	Difficult N = 16	F	Significance	95% Cis	Effect Size
	M (SD)	M (SD)				Hedges’ g
Secure	4.52 (1.85)	3.50 (1.37)	2.32	0.13	3.66–4.66	0.57
Fearful	3.18 (2.02)	5.06 (1.61)	8.53	0.004	3.70–4.65	0.96
Preoccupied	2.56 (1.58)	2.81 (2.04)	0.01	0.98	3.04–4.02	0.15
Dismissive	3.77 (1.72)	4.06 (1.95)	0.27	0.61	3.65–4.65	0.17
Overall Health	73.52 (18.81)	56.87 (20.24)	8.87	0.004	53.05–65.31	0.87

Note: MANOVA Wilk’s Lambda F(9,91) = 4.05, *p* = 0.002.

## Data Availability

The raw data supporting the conclusions of this article will be made available by the authors upon request.

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
