# Peer review of "Difficult Doctor–Patient Relationships and Attachment Style in Primary Care"

_healthcare, 2025, doi:10.3390/healthcare13222952_

Round 1

Reviewer 1 Report

Comments and Suggestions for Authors
  1. In the introduction section, the literature review could better synthesize gaps—for instance, emphasizing why this pilot using archival data adds value to the existing evidence. Some references are older (1990s, early 2000s); consider citing more recent studies (post-2020) on attachment in primary care or telehealth contexts to strengthen currency. Clarify whether “difficult patients” is being conceptualized as a behavioral construct, perception, or interactional outcome.
  2. The use of archival data (from 2016) should be more explicitly justified—why it remains relevant to 2025 practice, beyond being a pilot.
    Mentioning whether physicians were blinded to patient attachment results would clarify potential bias.
  3. Clarify the cut-off criteria for “difficult patients” (e.g., justification for DDPRQ-10 ≥ 30).

    Specify how missing data (if any) were handled.

    State whether statistical assumptions for MANCOVA (e.g., homogeneity of variance-covariance) were checked.

    Consider providing a power analysis or effect size estimates to contextualize sample adequacy (N=100).

  4. Include confidence intervals or effect sizes for key findings (e.g., fearful attachment, health ratings).
    Indicate sample size per analysis in the table captions for clarity
  5. In conclusion, avoid overgeneralizing beyond the study’s pilot and correlational nature.
    The discussion could benefit from a clearer distinction between clinical interpretation and public health implications.
    More explicit acknowledgment of directionality limitations (i.e., whether physician perception could influence difficulty ratings)
  6. Consider adding a conceptual figure (e.g., theoretical model linking attachment style → perceived difficulty → health outcome).

Reviewer 2 Report

Comments and Suggestions for Authors

I would like to thank for the opportunity to revise this manuscript, and the authors for their work.

This is an interesting paper, which aims to deepen the relationship between Doctor-Patient relationship and attachment style.

Before the publication, I have some minor suggestions.

1) Abstract: I would remove the references in line 10, and add a brief explanation of the adopted study design among the methods.

2) Introduction, lines 27-37. This section could benefit of some references

3) Materials and Methods: I suggest to specify how the sample size was calculated, and a paragraph on the adopted statistical analysis

4) Results: I would add a table regarding the descriptive analysis of sample characteristics. Maybe with the overall description in one column, and other two columns with the sample divided ("difficult" and "not difficult")?

Author Response

Dear Editor,

We appreciate the thoughtful suggestions you have made regarding the manuscript.

Our group has responded to your suggested changes and believe that the

manuscript is in better shape because of them.

With gratitude,

John Porcerelli, Reem Eissa, and Pierre Morris  

  • In the introduction section, the literature review could better synthesize gaps—for instance, emphasizing why this pilot using archival data adds value to the existing evidence. RESPONE: At the end of the Participants and Procedures paragraph, a rationale for using archival data was added. (Lines 114-116)

Some references are older (1990s, early 2000s); consider citing more recent studies (post-2020) on attachment in primary care or telehealth contexts to strengthen currency. RESPONSE: Two more recent studies were added to 1st paragraph (Mota et al, 2019) and 2nd paragraph (Porcerelli et al., 2024). Unfortunately, there are no recent studies to cite involving Dr-Pt relationships and attachment in primary care. (Line 28 and Line 43)

Clarify whether “difficult patients” is being conceptualized as a behavioral construct, perception, or interactional outcome. RESPONSE: In the first sentence of the DDPRQ-10 description states that the assessment of “difficult patient” is based on physician perception. (Lines 133-134)

  • The use of archival data (from 2016) should be more explicitly justified—why it remains relevant to 2025 practice, beyond being a pilot. RESPONE: At the end of the Participants and Procedures paragraph, a rationale for using archival data was added. (Lines 114-116)

               Mentioning whether physicians were blinded to patient attachment results would clarify potential bias. RESPONSE: The following statement was added to the   Participants and Procedures section: “Physicians were blind to the patient’s           attachment classification.” (Lines 109-110)

  • Clarify the cut-off criteria for “difficult patients” (e.g., justification for DDPRQ-10 ≥ 30). RESPONSE: 30 was the original cut-off score derived from the initial validation study. The reference was added to the end of the sentence. (Line 143)

               Specify how missing data (if any) were handled. RESPONSE: Date were complete –      no missing data.  A statement was added to the Participants & Procedures section. (Line 110-111)

               State whether statistical assumptions for MANCOVA (e.g., homogeneity of variance-          covariance) were checked. RESPONE: Box’s M results were added to the 4th           paragraph of the results section. (Lines 168-169)

Consider providing a power analysis or effect size estimates to contextualize sample adequacy (N=100). RESPONSE: Effect sizes (Hedges’ g) were added to Table 3.

  • Include confidence intervals or effect sizes for key findings (e.g., fearful attachment, health ratings). RESPONSE: Confidence intervals have been added to Table 3.

               Indicate sample size per analysis in the table captions for clarity. RESPONSE:    Sample Size added to Table 2.

  • In conclusion, avoid overgeneralizing beyond the study’s pilot and correlational nature. RESPONSE: The last sentence of the Conclusion section now reads “Knowing how to assess and manage encounters with such patients may improve …” (Lines 268-270)

The discussion could benefit from a clearer distinction between clinical                interpretation and public health implications. RESPONSE: The following sentence          was added to the end of the 3rd paragraph in the Discussion section: “It would be important to determine the prevalence of patients who avoid care following difficult           encounters to underscore the public health implications of such encounters.”  (Lines 242-       244)

               More explicit acknowledgment of directionality limitations (i.e., whether physician         perception could influence difficulty ratings). RESPONSE: Last sentence of the 4        paragraph of the Discussion section reads: “Future studies could include observer-          ratings of doctor-patient encounters to understand physicians’ contributions to    experiencing encounters as difficult.” (Lines 249-250)

  • Consider adding a conceptual figure (e.g., theoretical model linking attachment style → perceived difficulty → health outcome). RESPONSE: With all due respect, our group feels like such a figure would go beyond the present study. However, such a figure would be quite useful in a study that includes physician contributions to difficult encounters.